# Piperine Provides Neuroprotection against Kainic Acid-Induced Neurotoxicity via Maintaining NGF Signalling Pathway

**DOI:** 10.3390/molecules27092638

**Published:** 2022-04-20

**Authors:** Ting-Yang Hsieh, Yi Chang, Su-Jane Wang

**Affiliations:** 1P.H.D. Program in Nutrition & Food Science, College of Human Ecology, Fu Jen Catholic University, New Taipei City 24205, Taiwan; david711young@gmail.com; 2Department of Anesthesiology, Shin Kong Wu Ho-Su Memorial Hospital, Taipei 11101, Taiwan; m004003@ms.skh.org.tw; 3School of Medicine, Fu Jen Catholic University, New Taipei City 24205, Taiwan; 4Research Center for Chinese Herbal Medicine, College of Human Ecology, Chang Gung University of Science and Technology, Taoyuan 33303, Taiwan

**Keywords:** piperine, glutamate excitotoxicity, NGF, neuroprotection, kainic acid, hippocampus

## Abstract

The neuroprotective properties of piperine, the major alkaloid extracted from black pepper, have been under investigation, but its mechanism of action in excitotoxicity is still poorly understood. This study aimed to evaluate the protective effects of piperine with a focus on nerve growth factor (NGF) signalling in a kainic acid (KA) rat model of excitotoxicity. Rats were administered intraperitoneally (i.p.) piperine (10 or 50 mg/kg) before KA injection (15 mg/kg, i.p.). Our results show that KA exposure in rats caused seizure behaviour, intrinsic neuronal hyperactivity, glutamate elevation, hippocampal neuronal damage, and cognitive impairment. These KA-induced alterations could be restored to the normal state by piperine treatment. In addition, piperine decreased the expression of the NGF precursor proNGF and NGF-degrading protease matrix metalloproteinase 9, whereas it increased the expression of proNGF processing enzyme matrix metalloproteinase 7, NGF, and NGF-activated receptor TrkA in the hippocampus of KA-treated rats. Furthermore, KA decreased phosphorylation of the protein kinase B (Akt) and glycogen synthase kinase 3β (GSK3β) in the hippocampus, and piperine reversed these changes. Our data suggest that piperine protects hippocampal neurons against KA-induced excitotoxicity by upregulating the NGF/TrkA/Akt/GSK3β signalling pathways.

## 1. Introduction

Glutamate is the major excitatory neurotransmitter in the central nervous system (CNS) and plays a crucial role in neurological processes, including cognition, learning, and memory [1]. However, excessive activation of glutamate receptors under pathological conditions leads to Ca^2+^ overload and neuronal death [1,2]. Glutamate-induced excitotoxicity is considered a common hallmark of many neurological disorders [3,4] and has been linked to alterations in the expression of the neurotrophin nerve growth factor (NGF) [5,6]. NGF regulates neuronal survival by binding to TrkA, which activates several signalling pathways, such as protein kinase B (Akt) and glycogen synthase kinase 3β (GSK3β) [7,8]. Decreased NGF that accompanies neuronal death has been observed in the brains of excitotoxic injuries, and neuronal cell death from excitotoxicity also appears to be prevented by increasing NGF [5,6,9]. Importantly, the NGF pathway has emerged as a pharmacological target for neuroprotection and repair against excitotoxic brain injury or neurodegenerative disorders [10,11,12,13,14].

The search for new drugs that maintain glutamate homeostasis and regulate the cascades of intracellular signalling that can lead to neuronal death is very important. In this sense, plant-derived substances have been shown to be efficient in increasing NGF expression and protecting against glutamate excitotoxicity [9,15,16]. Piperine is an alkaloid extracted from black pepper, the chemical formula of C_17_H_19_NO_3_ (Figure 1A). However, this substance has poor solubility in water and low bioavailability [17,18]. Piperine has antioxidant, anti-inflammatory, and antitumour effects. In addition, piperine has been found to provide neural protective actions in various in vitro and in vivo experimental models [17,19,20]. It has been demonstrated, for example, that piperine can penetrate the blood–brain barrier (BBB) [18,21], protect against ischaemic brain injury and kainate-induced seizures or neurotoxicity [22,23,24], attenuate β-amyloid or oxidative stress-induced neuronal cell damage and death [25,26,27], and improve depression-like behaviour and cognitive impairment [26,28,29]. Despite the compelling evidence for the neuroprotective effects of piperine, its mechanism of action is not fully clarified. The aim of the present study was to investigate the protective effect of piperine with a focus on NGF signalling in an excitotoxicity rat model induced by kainic acid (KA), which is a glutamate analogue. Here, we propose that the neuroprotective mechanisms of piperine are associated with upregulation of the NGF signalling cascade.

## 2. Results

### 2.1. Piperine Pretreatment Attenuated Seizures and Hippocampal Glutamate Level Elevation in KA-Treated Rats

Systemic administration of KA to rodents provokes endogenous glutamate release and results in excitotoxic events, such as seizure activity [30]. To evaluate the protective effect of piperine on the seizure activity induced by KA, rats were intraperitoneally (i.p.) injected with piperine (10 or 50 mg/kg) 30 min before KA administration (15 mg/kg, i.p.), and the epileptic burst was detected by electroencephalogram (EEG) recording. The details of the experimental timeline are shown in Figure 1B. In Figure 2A, no epileptic bursts were observed in the control group. Compared with the control group, the KA-treated group exhibited a significant increase in the number of epileptic bursts. This enhancement was prevented in piperine + KA-treated rats compared with the KA group (Figure 2A). In the piperine (10 and 50 mg/kg) + KA group, quantification of the results showed a significant decrease in the number and duration of the epileptic bursts (F(2, 6) = 43.52, *p* < 0.001 vs. the KA group and F(2, 6) = 72.62, *p* < 0.001 vs. the KA group, respectively, Figure 2B). Obvious seizure behaviours were also observed in KA-injected rats. Piperine pretreatment (10 or 50 mg/kg) significantly decreased the severity of seizures over 180 min compared with the KA group (at 180 min: F(2, 24) = 28.3, *p* < 0.001, Figure 3A). In addition, we collected cerebrospinal fluid (CSF) to examine the glutamate level changes 3 days after treatment by using HPLC. As shown in Figure 3B, the extracellular glutamate levels were dramatically increased in the KA group (F(3, 16) = 304.7, *p* < 0.001 vs. the control group). Piperine pretreatment (10 or 50 mg/kg) significantly decreased the levels of glutamate compared with the KA group (*p* < 0.001, Figure 3B).

### 2.2. Piperine Pretreatment Attenuated KA-Induced Neuronal Cell Death in the Hippocampus of Rats

Three days after KA injection, brain tissue sections were stained with the neuronal marker neuronal nuclei (NeuN) and Fluoro-Jade B (FJB, identifying degenerating neurons) to study the neuroprotective effect of piperine on hippocampal neurons. As illustrated in Figure 4A, neuronal loss and neuronal degeneration were observed in the hippocampal CA1 and CA3 regions of the KA-injected rats, and piperine pretreatment (10 or 50 mg/kg) alleviated these KA-induced changes. The quantification showed that a significant reduction in the number of NeuN^+^ cells and a significant increase in the number of FJB^+^ cells were observed in the hippocampal CA1 and CA3 regions of the KA-injected rats (NeuN CA1: F(3, 12) = 119.4; NeuN CA3: F(3, 12) = 139.5; FJB CA1: F(3, 12) = 179.9; FJB CA3: F(3, 12) = 361.5, *p* < 0.0001 vs. the control group, Figure 4B). However, the number of NeuN^+^ cells in the hippocampal CA1 and CA3 regions was significantly increased in the piperine (10 and 50 mg/kg) + KA group compared with the KA group (CA1: *p* < 0.0001; CA3: *p* < 0.0001, Figure 4B). Similarly, the number of FJB^+^ cells in the hippocampal CA1 and CA3 regions was significantly reduced in the piperine (10 and 50 mg/kg) + KA group compared with the KA group (CA1: *p* < 0.0001; CA3: *p* < 0.0001, Figure 4C).

### 2.3. Piperine Pretreatment Maintained the Levels of NGF in the Hippocampus of Rats with KA Injection

As NGF has a critical function in neuronal survival during brain injury induced by excitotoxicity [6,9], we assayed the expression of NGF and its precursor proNGF in the hippocampus of KA-treated rats. As depicted in Figure 5, KA significantly decreased the levels of NGF and significantly increased the proNGF levels compared with the control group (NGF: F(3, 16) = 17.5; proNGF: F(3, 16) = 31.6, *p* < 0.001). However, these alterations in NGF and proNGF expression were reversed by piperine pretreatment (*p* < 0.001). In addition, a significant decrease in the level of matrix metalloproteinase 7 (MMP7, a protease converting proNGF to NGF) and a marked increase in the level of matrix metalloproteinase 9 (MMP9, an NGF degrading enzyme) were observed in the hippocampus of KA-treated rats (MMP7: F(3, 16) = 43.5; MMP9: F(3, 16) = 80.7, *p* < 0.001 vs. the control group; Figure 5). Pretreatment with piperine (10 or 50 mg/kg) effectively restored the protein levels of MMP7 (*p* < 0.001) and MMP9 (*p* < 0.001) to the control levels in the hippocampus of the rats (Figure 5).

### 2.4. Piperine Pretreatment Increased the Phosphorylation Levels of TrkA, Akt, and GSK3β in the Hippocampus of KA-Treated Rats

To further investigate the possible molecular mechanisms of piperine, we analysed the high-affinity membrane receptor tyrosine kinase A (TrkA) of NGF and its downstream Akt/GSK3β signalling pathway [7,8]. As shown in Figure 6, KA decreased the TrkA expression (TrkA: F(3, 16) = 31.39, *p* < 0.001 vs. the control group) and significantly reduced the phosphorylation levels of TrkA, Akt and GSK3β compared to the control group (pTrkA: F(3, 16) = 18.51; pAkt: F(3, 16) = 36.69; pGSK3β: F(3, 16) = 404.3, *p* < 0.001). These changes induced by KA were significantly increased in the piperine (10 and 50 mg/kg) + KA group compared with the KA group (*p* < 0.001).

### 2.5. Piperine Pretreatment Improved KA-Induced Spatial Memory Dysfunction in Rats

To determine the effect of piperine on spatial learning and memory, we analysed the rat performance in the MWM test. As shown in Figure 7A, the swimming tracks of the rats in each group were recorded. Figure 7B shows that KA significantly increased the time to find the platform (escape latency) compared with the control group (F(3.16) = 252.37, *p* < 0.001). Pretreatment with piperine (10 and 50 mg/kg) effectively decreased the escape latency compared with the KA group (*p* < 0.001). Similarly, the swimming distance of the KA group markedly increased compared with that of the control group (F(3.16) = 68.7, *p* < 0.001), which was shortened by piperine pretreatment (*p* < 0.001, Figure 7B).

## 3. Discussion

The pathophysiology of numerous neurological diseases has been associated with glutamatergic excitotoxicity [4,16,31]. Thus, research on new neuroprotective strategies with the ability to mitigate glutamate toxicity has received increasing attention. Given this background, we evaluated the effect of piperine in an excitotoxicity rat model induced by systemic KA administration, investigating its possible mechanism of action. This study provided evidence that piperine prevents KA-induced excitotoxicity, which could result from activating TrkA/Akt/GSK3β signalling by increasing NGF expression.

KA, an analogue of glutamate, can increase glutamate levels and glutamate receptor overactivation, which leads to an epileptic burst and excitotoxic neuronal death [31,32]. In the current study, we found that pretreatment with piperine significantly attenuated KA-induced abnormal electrical brain activity and seizure behaviour, indicating that piperine exerts an anticonvulsant effect, which is consistent with previous studies [22,23,33]. In addition, we found that piperine pretreatment reduced the KA-induced increase in glutamate concentrations and attenuated neuronal damage in the hippocampus. Thus, the beneficial effect of piperine on the generation of KA-induced seizures and neuronal damage can be at least partially ascribed to the inhibition of glutamate-mediated overexcitation. Although the exact mechanism of the anticonvulsant and neuroprotective effects of piperine remains to be further clarified, reducing glutamate release may provide some answers. For example, piperine has been found to suppress the release of glutamate by rat hippocampal nerve terminals [34]. Furthermore, the inhibitory effects of clinical antiepileptic drugs on glutamate release by human and rat brain tissues has been proposed [35,36]. Thus, we can infer that piperine pretreatment prevents the increase in glutamate release in the hippocampus of KA-treated rats, which may partially explain the neuroprotective effect of piperine against KA-induced hippocampal neuronal loss and, consequently, its anticonvulsant effect.

NGF, a member of the neurotrophin family of growth factors, is implicated in the regulation of neuronal survival, differentiation, and growth by binding to its high-affinity receptor (TrkA) [10,37]. Decreased NGF levels in the brain have been shown to be associated with KA-induced neuronal damage [5,6]. In this study, as expected, KA decreased NGF levels in the hippocampus, and this phenomenon was reversed by piperine pretreatment. Regarding the changes in the expression of NGF in the hippocampus, effects on NGF synthesis or NGF degradation might be involved [38,39,40]. Studies have already revealed that the NGF precursor proNGF is released when neuronal activity is activated and it is converted to NGF by MMP7; then, any NGF that does not bind to TrkA is rapidly degraded by MMP9 [41,42,43]. In the present study, we found that KA decreased MMP7 expression, while it increased MMP9 expression. These KA-induced changes were reversed by piperine pretreatment. In addition, we found that the expression level of proNGF increased in the hippocampus of KA-treated rats, which is consistent with the findings of previous studies [44]. Piperine pretreatment was also able to prevent the KA-induced increase in proNGF levels. ProNGF is a potent apoptotic ligand that interacts with the p75 neurotrophin receptor (p75^NTR^) [45]. Increased proNGF that accompanies neuronal death has been observed in the hippocampus after seizure, and this excitotoxic neuronal loss also appears to be prevented by proNGF inhibition via increasing MMP7 activity [44,46]. We confer that i.p. administration of KA to rats decreased MMP7, leading to proNGF accumulation and neuronal death in the hippocampus. Piperine pretreatment might increase MMP7, as well as decrease proNGF, and in doing so, it could increase NGF to preserve KA-induced hippocampal neuronal damage. Overall, the neuroprotective effect of piperine against KA-induced hippocampal neuronal damage is attributable to its NGF enhancement, which is elicited via an increase in NGF synthesis and a decrease in NGF degradation.

Moreover, NGF exerts a protective effect against excitotoxicity-induced neuronal death by activating the TrkA signalling pathways, including the Akt/GSK3β pathway [7,8,47]. Inhibition of the Akt/GSK3β pathway has been shown to offset the ability of NGF to promote cell survival [14,48]. In our current study, we found that the activation of TrkA, Akt, and GSK3β in the hippocampus was decreased by KA administration, which was markedly reversed by piperine. Based on our data, we hypothesize that piperine exerts a protective effect against KA-induced hippocampal neuronal damage by activating the TrkA/Akt pathway through increasing NGF. In addition, dysregulation of NGF signalling has been shown to cause cognitive dysfunction [8,15,40,49,50]. A previous study also revealed that increased NGF and TrkA expression and activation of the Akt/GSK3β pathway could ameliorate brain hypoperfusion-induced cognitive impairment in rats [51]. In this study, we also evaluated memory function using the MWM test and found that piperine pretreatment improved memory deficits in KA-treated rats. Thus, we speculated that piperine could prevent hippocampal neuronal damage by increasing NGF-induced TrkA/Akt/GSK3β activation, thus contributing to the improvement in memory dysfunction in rats with KA-induced seizures. In fact, the beneficial effects of piperine on cognitive function have been proposed in various animal models of neurological disorders, including autism, Alzheimer’s disease, and Parkinson’s disease [52,53,54].

The ability of piperine to pass through the blood–brain barrier (BBB) has numerous beneficial effects on the CNS, which was revealed in in vivo studies [17,18,21]. For example, piperine was found to alleviate ischaemia-induced brain damage [24], to improve depression-like behaviour and cognitive dysfunction [29,52,54] and to promote neurogenesis [26]. Regarding the neuroprotective mechanisms of piperine, it has been reported that this beneficial effect is associated with influencing neurotransmitter concentrations [55] and suppressing oxidative stress [25] and inflammatory responses [23,33]. In addition to these possible mechanisms, our previous [34] and present study results presume that inhibiting glutamate release and its antiexcitotoxic properties may partly contribute to the neuroprotective effects of piperine in the brain. On the other hand, piperine has been shown to have an excellent safety profile [56,57]. For example, the oral administration (5–50 mg/kg/day) of piperine to rats for 90 days showed nontoxic effects [58]. In addition, the average daily consumption of piperine is approximately 14–54 mg/person/day [58]. Clinical trials also reported that oral administration of a single large piperine dose (45.7 mg/kg) produced health-promoting effects [59]. The doses of piperine (10 and 50 mg/kg) used in the present study are similar to those used in these studies, which indicates that the dose we used is safe and makes our results more relevant to clinical studies.

In the present study, although we observed significant anticonvulsant and neuroprotective effects of piperine against KA-induced seizures and excitotoxic injury, limitations exist. First, our findings indicate that the effects of piperine are related to NGF signalling upregulation in the KA seizure rat model, but the in-depth molecular mechanism that underlies its effects requires further verification. Second, the present study focused only on the effects of piperine on neurons. Because KA-induced seizures and excitotoxicity involve glial activation and suppressing glial activation protects against the neuronal damage caused by KA-induced seizure [60], further study is required to elucidate the effects of piperine on astrocytes/microglia in KA seizure rats. 

## 4. Materials and Methods

### 4.1. Animals and Experimental Design

Male Sprague–Dawley rats weighing 200–225 g were purchased from BioLASCO (Taipei, Taiwan) and maintained at the Animal Care Services of Fu Jen Catholic University, Taiwan. Rats were kept under standardized conditions (22–24 °C, 55% relative humidity, 12 h/12 h light/dark cycle) for 1 week, and food and water were available ad libitum. The experimental protocol was reviewed and approved by the Institutional Animal Care and Use Committee (IACUC) of Fu Jen Catholic University (17 December 2020; A10944).

The excitotoxicity animal model was induced by intraperitoneal (i.p.) administration of KA (K0250, Sigma-Aldrich, St. Louis, MO, USA; dissolved in normal saline) at a dose of 15 mg/kg. Rats were divided into 4 groups: the control group (i.p. injected DMSO alone, Sigma-Aldrich), KA group, piperine (P49007, Sigma-Aldrich) 10 mg/kg + KA group, and piperine 50 mg/kg + KA group. Piperine (dissolved in DMSO) was i.p. injected 30 min before KA injection. 

### 4.2. Seizure Score

After KA administration, spontaneous seizure behaviour of the rats was scored every 30 min for up to 3 h according to the modified Racine scale: (0) no response; (1) staring and nodding; (2) wet dog shakes (WDS), drooling, and pawing; (3) WDS, drooling, and forelimb clonus; (4) WDS, drooling, forelimb clonus with rearing and jumping; and (5) rearing, jumping, falling, status epilepticus, and even death [61].

### 4.3. EEG Recording

Rats were anaesthetized via inhalation of 3% sevoflurane (Baxter healthcare corporation, Guayama, Puerto Rico, USA). The rats were secured on a stereotaxic frame while a hot blanket was used to maintain their body temperature. After a central incision on the scalp was made, four burr holes were made in the skull according to the stereotaxic atlas (AP: +1.8 mm, ML: ±1.5 mm and AP: −3 mm, ML: ±2.5 mm relative to Bregma) using a hand-held dental drill (#8424, Pinnacle Technology, Inc., Lawrence, KS, USA) [62]. Then, stainless steel screws were drilled with wire leads (#8247, Pinnacle Technology, Inc., Lawrence, KS, USA) into these pilot holes. The EEG/EMG Rat Headmount (#8239, Pinnacle Technology, Inc., Lawrence, KS, USA) was implanted in the head, the EMG probes were inserted bilaterally into the nuchal muscles, and the probes were fixed to the skull with dental acrylic powder. EEG recordings were taken 7 days after the surgery and were monitored for 3 h after KA injection. The EEG equipment consisted of a 100× preamplifier (#8213, Pinnacle Technology, Inc., Lawrence, KS, USA) to monitor the data. The EEG data were collected from the data acquisition/conditioning system (#8206, Pinnacle Technology, Inc., Lawrence, KS, USA) and recorded with PAL-8200 software (Version 1. 6. 6, Pinnacle Technology, Lawrence, KS, USA). The spikes were analysed by Sirenia Seizure Pro software (Version 1. 6. 6, Pinnacle Technology, Lawrence, KS, USA).

### 4.4. Immunohistochemistry

Rats were transcardially perfused with 300 mL normal saline and 300 mL ice-cold 4% paraformaldehyde in phosphate-buffered saline (PBS, 0.1 M, pH 7.4, Sigma-Aldrich) under zoletil (40 mg/kg, i.p.; Virbac Lab, Carros, France) anaesthesia. Their brains were quickly removed and postfixed with 4% paraformaldehyde (Sigma-Aldrich) in PBS at 4 °C overnight. The brains were transferred to 30% sucrose (Sigma-Aldrich) in 0.1 M PBS at 4 °C. Coronal brain sections (20 µm) were then cut on a cryostat (Leica CM3050 S, Wetzlar, Germany).

Floating brain sections were blocked in PBS containing 5% normal goat serum (Abbkine, China) and 0.3% Triton X-100 (Sigma-Aldrich) for 1 h at room temperature. After blocking, the sections were incubated with the neuronal marker anti-NeuN (ab177487, Abcam, Cambridge, UK; diluted 1:500) for 60 min at room temperature followed by rinsing in PBS three times. Next, the sections were incubated with goat anti-rabbit antibody conjugated to DyLightTM 594 (35560, Invitrogen, IL, USA) in the dark for 90 min at room temperature. They were washed three times in PBS and stained with DAPI (D9542, Sigma-Aldrich, 1 µg/mL) for nuclear staining for 10 min at room temperature. After the final washes, the brain sections were mounted on gelatin-coated glass slides and allowed to air-dry in the dark.

The degeneration of the hippocampal neurons following KA-induced excitotoxicity was assessed with FJB (#1FJB, Histo-Chem, Jefferson, AR, USA) as described previously [63]. For FJB staining processing, the air-dried slides were rehydrated with decreasing concentrations of ethanol (100%, 95%, 70%) and immersed in distilled water. They were moved to 0.06% potassium permanganate (KMnO4, Sigma-Aldrich) and slowly shaken for 10 min, followed by rinsing in distilled water for 2 min. The slides were then soaked in 0.001% FJB solution containing 0.1% acetic acid with gentle shaking for 10 min and rinsed in distilled water three times for 1 min each. After air-drying in the dark, they were cleared by immersion in xylene for 2 min and coverslipped with Entellan^®^ (Sigma-Aldrich). The CA1 and CA3 regions of the hippocampus were visualized with an ImageXpress^®^ Micro confocal microscope (Molecular Devices, CA, USA) with a 40× objective. The numbers of NeuN^+^ cells (indicating surviving neurons) and FJB^+^ cells (indicating degenerative neurons) were counted in an area of 0.35 mm × 0.35 mm by ImageJ (version 1.43, Synoptics, Cambridge, UK) on 3 slices per animal.

### 4.5. Morris Water Maze

The rats were tested in the Morris water maze for spatial memory ability, as reported previously [64]. The water maze consisted of a black round pool (diameter: 150 cm; height: 70 cm) that was filled with tap water (23 ± 1 °C, depth: 40 cm) and divided into four quadrants by two imaginary lines crossing the pool centre. A transparent escape platform (diameter: 15 cm) was placed in the centre of the pool and submerged 1 cm below the water surface. Each rat was put into the pool facing the wall and trained to search for the hidden platform for a maximum of 2 min (4 times daily, from a different quadrant, with 5 min intervals) for 3 consecutive days. If the rat failed to find the hidden platform within 2 min, the examiner gently guided them to the platform where the animals were permitted to stay for 1 min. Three days after KA administration, the time and distance travelled to reach the hidden platform were recorded using a video tracking system (Version 3.0, Panlab, Barcelona, Spain).

### 4.6. Glutamate Levels 

Rats were anaesthetized with zoletil (40 mg/kg; i.p.), and cerebrospinal fluid (CSF, ~100 μL) was collected from the cisterna magna [65]. The CSF was filtered with a centrifugal filter (UFC5030, Merck KGaA, MA, USA) (10,000× *g* for 10 min at 4 °C) and frozen at −80 °C until use. The glutamate level was determined by injecting the filtered CSF into the HPLC system with electrochemical detection (HTEC-500, Eicom, Kyoto, Japan) according to the previously described method [66].

### 4.7. Western Blotting

After CSF collection, the rats were decapitated for hippocampal sampling. The hippocampus was quickly extracted and homogenized in ice-cold lysis buffer (Sigma-Aldrich) supplemented with phosphatase and protease inhibitors (Roche, Mannheim, Germany) as described previously [63]. Lysates were incubated on ice for 30 min and then centrifuged at 13,200 rpm for 10 min at 4 °C. The supernatants were harvested, and the protein concentration was determined by a bicinchoninic acid (BCA) kit (Thermo Scientific). Equal concentrations of protein (30 μg) from each group were electrophoresed in 12% SDS–PAGE and transferred to polyvinylidene fluoride (PVDF) membranes (GE Healthcare, Amersham, UK). After 1 h of blocking with 5% nonfat milk in Tris-buffered saline containing 0.1% Tween-20, the blots were incubated overnight at 4 °C with primary antibodies (β-actin, #3700, 1:8000; TrKA, #2505, 1:20,000; AKT, #9272, 1:4000; phospho-AKT, #9271, 1:4000; GSK3β, #9315, 1:8000; phospho-GSK3β, #9336, 1:2000; all from Cell Signaling Technology, Beverly, MA, USA; NGF, ab52918, 1:5000; ProNGF, ab68151, 1:15,000; MMP9, ab58803, 1:1000; all from Abcam, Cambridge, UK; MMP7, GTX104658, 1:500; from GeneTex, Irvine, CA, USA; phospho-TrKA, SC-8058, 1:500; from Santa Cruz Biotechnology, Dallas, TX, USA). After washing with TBST (0.1% Tween-20 in Tris-buffered saline), the membranes were incubated with goat anti-rabbit or anti-mouse HRP-linked secondary antibody (1:5000; GeneTex) in 5% nonfat milk for 1 h at room temperature. The immunoreactive bands on the membrane were detected with a chemiluminescent kit (GE Healthcare) and exposed to X-ray film (GE Healthcare). The optical density of the protein bands was quantified using ImageJ software (version 1.43, Synoptics, Cambridge, UK), normalized to β-actin, and presented as a relative value.

### 4.8. Statistical Analysis

All statistical analyses were carried out with SPSS 18 software, and all figures were made using GraphPad Prism 8 software. Values are expressed as the means ± SEM of three or more independent experiments. One-way ANOVA followed by the Scheffé post hoc test was used to assess group differences. The results were defined as statistically significant when *p* < 0.05.

## 5. Conclusions

In conclusion, the present study demonstrates that piperine attenuates excitotoxic damage in a rat model induced by KA. This neuroprotective effect of piperine may result from attenuating glutamate hyperactivity and upregulating the NGF-mediated TrkA/Akt/GSK3β signalling pathway in the hippocampus (Figure 8). The results from this study provide a new framework for the neuroprotective mechanisms of piperine, suggesting that piperine is likely to be a promising therapy for preventing excitotoxicity-related brain diseases.

## Figures and Tables

**Figure 1 molecules-27-02638-f001:**
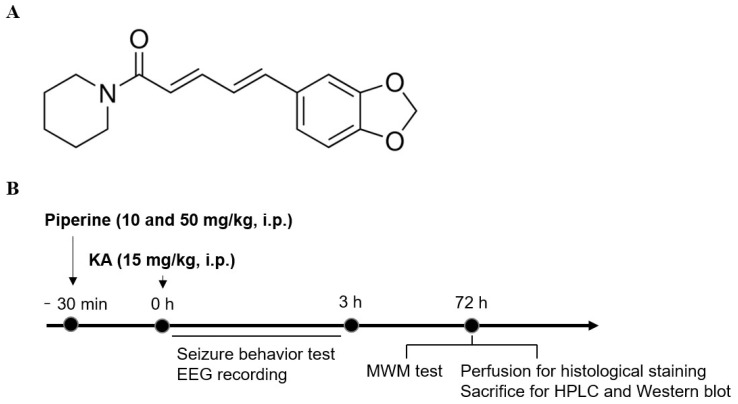
(**A**) Chemical structure of piperine. (**B**) Experimental design. Rats were administered piperine (10 and 50 mg/kg, i.p.) 30 min before KA injection (15 mg/kg, i.p.). Seizure behaviour and epileptic bursts were recorded 3 h after KA administration. At 3 days after KA injection, the rats were examined by the Morris water maze (MWM) test and sacrificed for high-performance liquid chromatography (HPLC), Western blot, and immunohistochemistry (IHC) studies.

**Figure 2 molecules-27-02638-f002:**
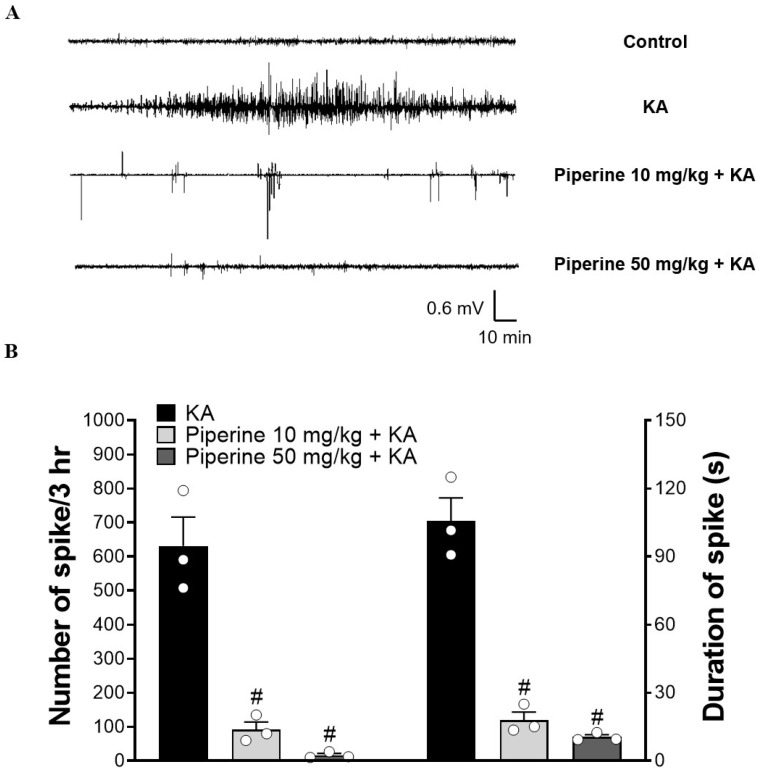
Effect of piperine pretreatment on electrographic seizures in rats with KA. (**A**) Representative EEG traces show epileptic bursts within 3 h. (**B**) The epileptic bursts are presented as the mean number and mean duration of seizure spikes. Data are presented as the means ± S.E.M. (n = 3). # *p* < 0.001 versus the KA group (ANOVA/Scheffé).

**Figure 3 molecules-27-02638-f003:**
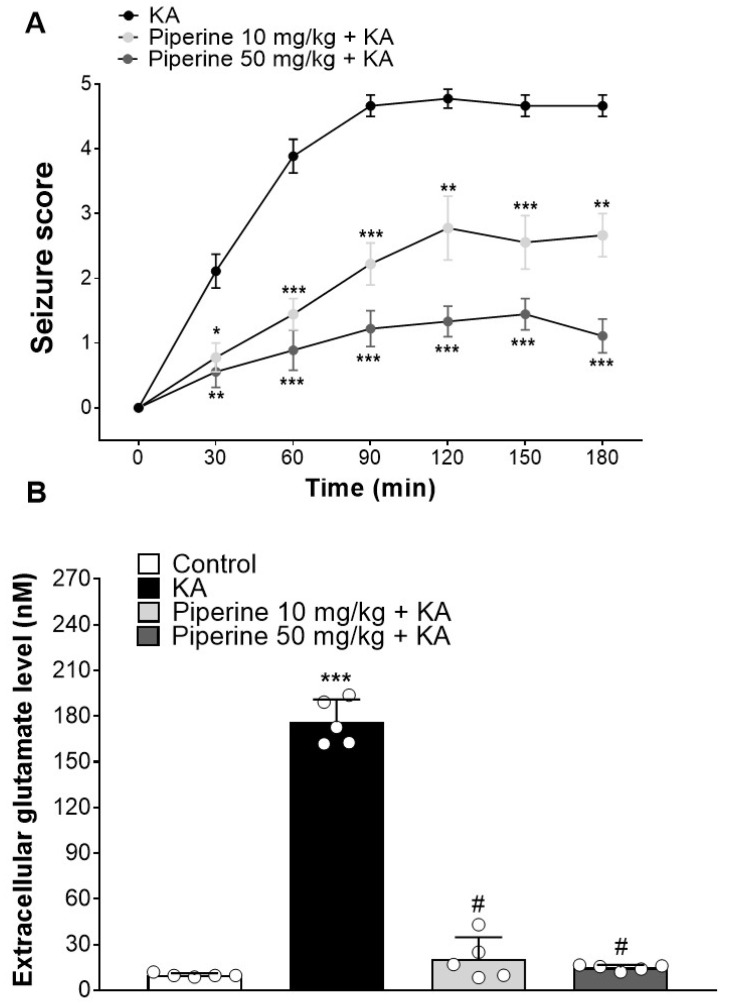
Effect of piperine pretreatment on seizure severity and extracellular glutamate levels in rats with KA. (**A**) Seizure scores are recorded every 30 min (n = 9). * *p* < 0.05; ** *p* < 0.01; *** *p* < 0.001 versus the KA group. (**B**) The extracellular glutamate levels from the cisterna magna were evaluated by HPLC (n = 5). Data are presented as the means ± S.E.M. # *p* < 0.001 versus the KA group; *** *p* < 0.001 versus the control group (ANOVA/Scheffé).

**Figure 4 molecules-27-02638-f004:**
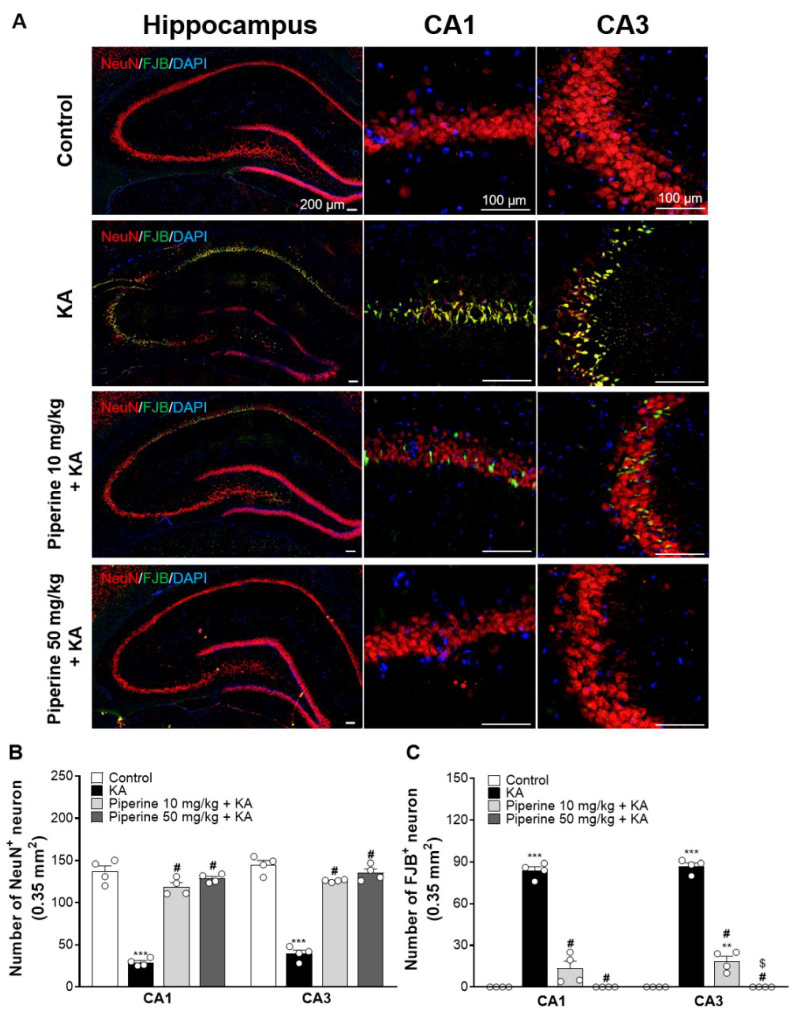
Effect of piperine pretreatment on neuronal damage in the hippocampus of KA-treated rats. (**A**) CA1 and CA3 regions of the hippocampus are observed with NeuN, FJB, and DAPI immunofluorescent staining. Scar bar for hippocampus = 200 µm; scar bar for CA1 and CA3 regions = 100 µm. (**B**,**C**) Quantitative analysis of NeuN-positive and FJB-positive neurons in the hippocampal CA1 and CA3 regions. Data are presented as the means ± S.E.M. (n = 4). # *p* < 0.001 versus the KA group; ** *p* < 0.01 versus the control group; *** *p* < 0.001 versus the control group; ^$^ *p* < 0.01 versus the KA + piperine 10 mg/kg group (ANOVA/Scheffé).

**Figure 5 molecules-27-02638-f005:**
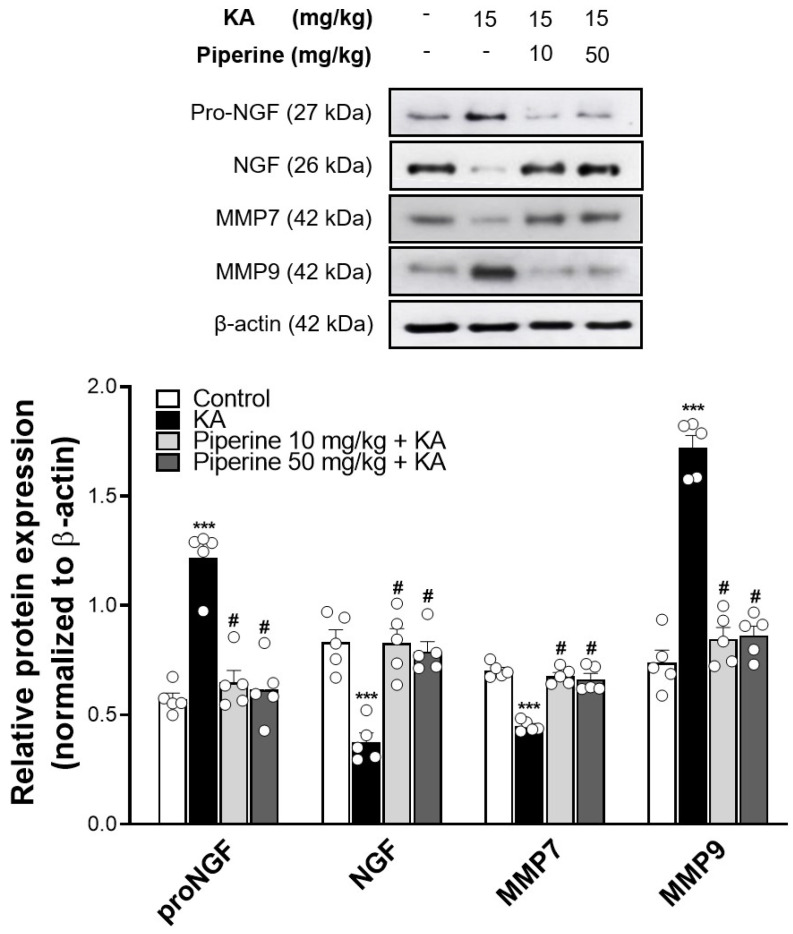
Effect of piperine pretreatment on pro-NGF, NGF, MMP7, and MMP9 protein levels in the hippocampus of KA-treated rats. The upper panels show representative Western blots. The lower panel is the relative density bar graph. Quantification of each protein was normalized to β-actin. Data are presented as the means ± S.E.M. (n = 5). # *p* < 0.001 versus the KA group; *** *p* < 0.001 versus the control group (ANOVA/Scheffé).

**Figure 6 molecules-27-02638-f006:**
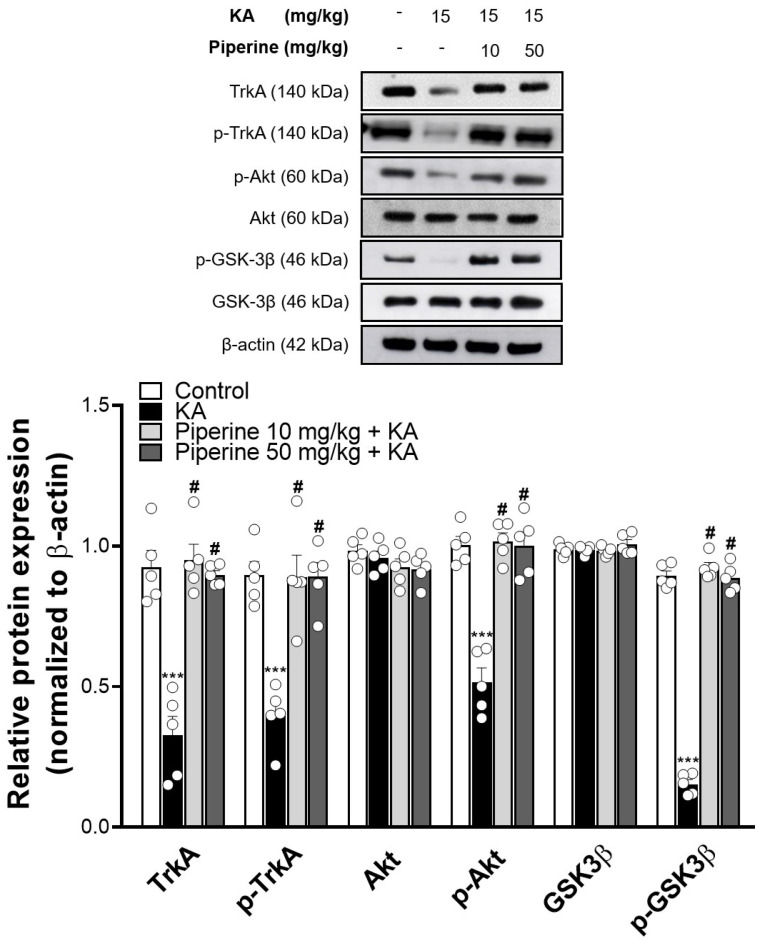
Effect of piperine pretreatment on TrkA, p-TrkA, Akt, p-Akt, GSK3β, and p-GSK3β protein levels in the hippocampus of KA-treated rats. The upper panel shows a representative Western blot. The lower panel is the relative density bar graph. Quantification of each protein was normalized to β-actin. Data are presented as the means ± S.E.M. (n = 5). # *p* < 0.001 versus the KA group; *** *p* < 0.001 versus the control group (ANOVA/Scheffé).

**Figure 7 molecules-27-02638-f007:**
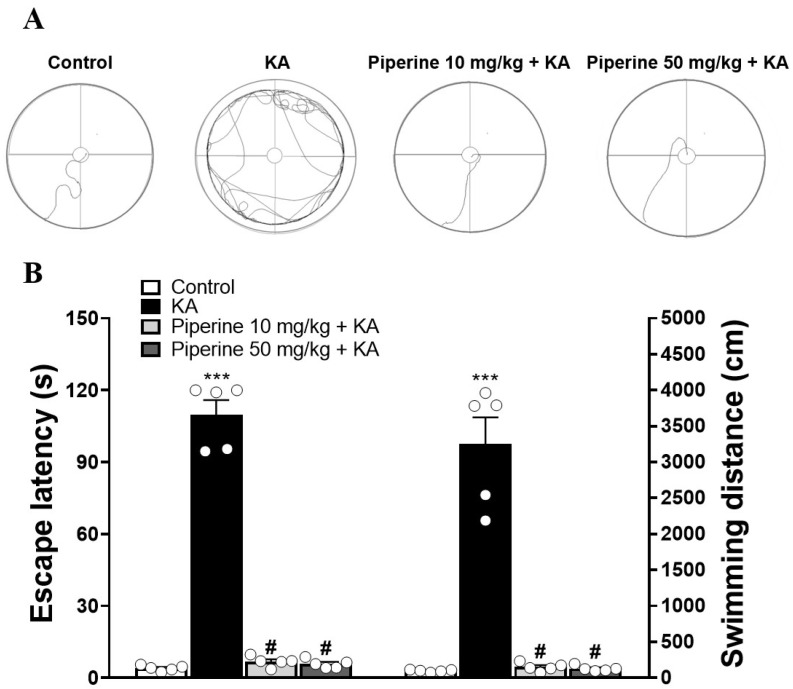
Effect of piperine pretreatment on spatial memory deficits in rats with KA. (**A**) The swimming path of rats in the MWM test. (**B**) Quantitative data of memory deficits are presented in the mean escape latency and mean swimming length. Data are presented as the means ± S.E.M. (n = 5). # *p* < 0.001 versus the KA group; *** *p* < 0.001 versus the control group (ANOVA/Scheffé).

**Figure 8 molecules-27-02638-f008:**
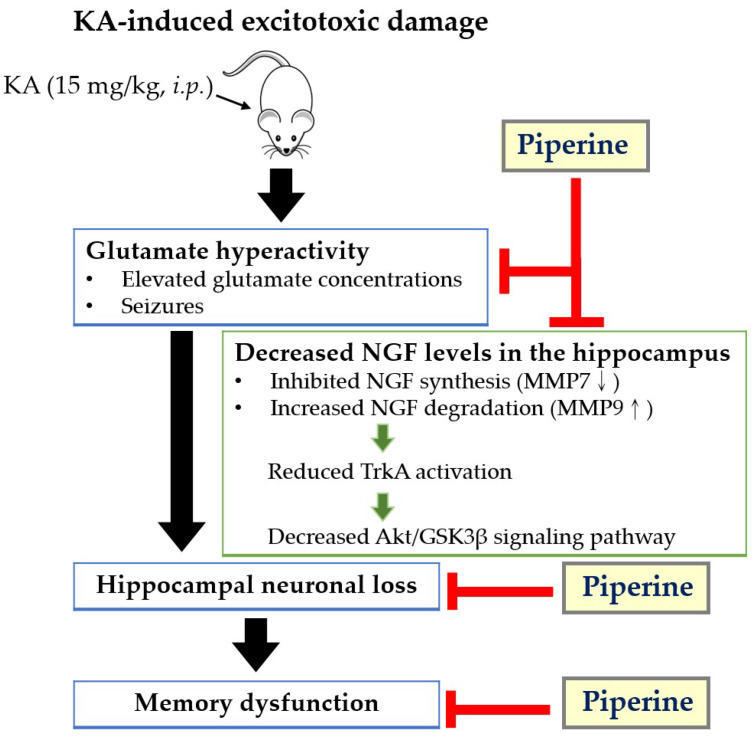
Schematic diagram representing the neuroprotective effects of piperine in the hippocampus of rats induced by KA. Piperine pretreatment ameliorates glutamate hyperactivity, upregulates NGF expression by increasing MMP7 levels and decreasing MMP9 levels, and enhances TrkA and Akt/GSK3β activation, which prevents KA-induced neuronal loss and maintains memory function. Arrows indicate KA-induced regulation, and T-bars indicate piperine-mediated inhibitory effects.

## Data Availability

The data presented in this study are available on request from the corresponding author.

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
