# Peer review of "Piperine Provides Neuroprotection against Kainic Acid-Induced Neurotoxicity via Maintaining NGF Signalling Pathway"

_molecules, 2022, doi:10.3390/molecules27092638_

Round 1
Reviewer 1 Report
After carefully reading the manuscript, I conclude that the abstract, introduction, and other chapters cover the issues discussed in an extensive and proper manner. The conclusions presented by the authors are consistent with the evidence and relate to the main research issue.
The weakness of the reviewed work concern as follows:
- As the name of the journal suggests, it is a journal dedicated to molecules. Therefore, it seems necessary to supplement this manuscript with the information about piperine and kainic acid molecules. I suggest that introduction section should be improved by including the chemical background of discussed molecules (e.g. chemical properties such as: pKa, log P, etc). Additionally, due to the fact that these compounds: piperine and kainic acid show opposite biological effects, it seems interesting to compare their physicochemical properties. I suggest to supplement this information in the form of a short paragraph or table within the introduction section.
- A significant drawback of the work is the lack of a concise description of the limitations of the study, e.g. in the form of an additional chapter.
I recommend publication after minor revision.
Author Response
We thank the reviewer for the critical comments and constructive suggestions.
- As the name of the journal suggests, it is a journal dedicated to molecules. Therefore, it seems necessary to supplement this manuscript with the information about piperine and kainic acid molecules. I suggest that introduction section should be improved by including the chemical background of discussed molecules (e.g. chemical properties such as: pKa, log P, etc). Additionally, due to the fact that these compounds: piperine and kainic acid show opposite biological effects, it seems interesting to compare their physicochemical properties. I suggest to supplement this information in the form of a short paragraph or table within the introduction section.
As suggestion by the reviewer, the sentence is modified to「Piperine is an alkaloid extracted from black pepper, the chemical formula of C17H19NO3 (Figure 1A). However, this substance has poor solubility in water and low bioavailability」(Page 2, Lines 47-49).
- A significant drawback of the work is the lack of a concise description of the limitations of the study, e.g. in the form of an additional chapter.
As suggestion by the reviewer, the sentences「In the present study, although we observed significant anticonvulsant and neuroprotective effects of piperine against KA-induced seizures and excitotoxic injury, limitations exist. First, our findings indicate that the effects of piperine are related to NGF signaling upregulation in the KA seizure rat model, but the in-depth molecular mechanism that underlies its effects requires further verification. Second, the present study focused only on the effects of piperine on neurons. Because KA-induced seizures and excitotoxicity involve glial activation and suppressing glial activation protects against the neuronal damage caused by KA-induced seizure [60], further study is required to elucidate the effects of piperine on astrocytes/microglia in KA seizure rats.」are added in the discussion section (Page 10, Lines 272-279).
Reviewer 2 Report
This manuscript has clearly demonstrated that pre-administration of piperine suppressed KA-induced excitotoxicity via NGF/TrkA/Akt/GSK3β signaling pathway. The descriptive data are well-demonstrated by the biochemical methods as well as EGG and MWM tests. However, I think there are a few points that need to be improved.
- Pre-administration of peperine suppresses the abnormal spikes of EGG that begin immediately after KA injection, which appear to be earlier than activation of NGF signaling pathway. The authors previously reported that piperine suppressed the Glu release in rat hippocampus. Please cite the paper and discuss the immediately effects of piperine.
- In the Discussion, the authors described that KA increased proNGF expression. Please discuss and describe why KA increased the expression of proNGF by referring to the literature. For example, Ref. 5 states that intraperitoneal administration of KA to rats caused an increase in NGF mRNA and a decrease in protein levels in the hippocampus.
- In Figure 2. A, the scale bar in this figure indicates 20 min/4 mm, which gives these EGGs 6.5 hours. Please check it.
- In Figure 3. A, since the maximum value of seizure score is 5, so I think it is better not to have “6” on the vertical axis.
- In Figure 7. B, the authors are experimented with spatial memory using the MWM test. Please describe what evaluation each of the “escape latency” and the “swimming distance” is used for.
Author Response
We thank the reviewer for the critical comments and constructive suggestions.
- Pre-administration of peperine suppresses the abnormal spikes of EGG that begin immediately after KA injection, which appear to be earlier than activation of NGF signaling pathway. The authors previously reported that piperine suppressed the Glu release in rat hippocampus. Please cite the paper and discuss the immediately effects of piperine.
As suggestion by the reviewer, the paper is cited (Reference 34) and the sentence 「Thus, we can infer that piperine pretreatment prevents the increase in glutamate release in the hippocampus of KA-treated rats, which may partially explain the neuroprotective effect of piperine against KA-induced hippocampal neuronal loss and, consequently, its anticonvulsant effect.」is added in the discussion section (Page 8, Lines 207-210).
- In the Discussion, the authors described that KA increased proNGF expression. Please discuss and describe why KA increased the expression of proNGF by referring to the literature. For example, Ref. 5 states that intraperitoneal administration of KA to rats caused an increase in NGF mRNA and a decrease in protein levels in the hippocampus.
As suggestion by the reviewer, the sentences 「In addition, we found that the expression level of proNGF increased in the hippocampus of KA-treated rats, which is consistent with the findings of previous studies [44]. Piperine pretreatment was also able to prevent the KA-induced increase in proNGF levels. ProNGF is a potent apoptotic ligand that interacts with the p75 neurotrophin receptor (p75NTR) [45]. Increased proNGF that accompanies neuronal death has been observed in the hippocampus after seizure, and this excitotoxic neuronal loss also appears to be prevented by proNGF inhibition via increasing MMP7 activity [44,46]. We confer that i.p. administration of KA to rats decreased MMP7, leading to proNGF accumulation and neuronal death in the hippocampus. Piperine pretreatment might increase MMP7, as well as decrease proNGF, and in doing so, it could increase NGF to preserve KA-induced hippocampal neuronal damage. Overall, the neuroprotective effect of piperine against KA-induced hippocampal neuronal damage is attributable to its NGF enhancement, which is elicited via an increase in NGF synthesis and a decrease in NGF degradation.」are added in the discussion section (Page 9, Lines 222-235).
- In Figure 2. A, the scale bar in this figure indicates 20 min/4 mm, which gives these EGGs 6.5 hours. Please check it.
As suggestion by the reviewer, Figure 2A is modified.
- In Figure 3. A, since the maximum value of seizure score is 5, so I think it is better not to have “6” on the vertical axis.
As suggestion by the reviewer, Figure 3A is modified.
- In Figure 7. B, the authors are experimented with spatial memory using the MWM test. Please describe what evaluation each of the “escape latency” and the “swimming distance” is used for.
As suggestion by the reviewer, the sentence is modified to 「To determine the effect of piperine on spatial learning and memory, we analyzed the rat performance in the MWM test (Page 7, Lines 172-173). Three days after KA administration, the time and distance traveled to reach the hidden platform were recorded using a video-tracking system (Page 11, Lines 360-362)」.
Reviewer 3 Report
Title of the study:Piperine provides neuroprotection against kainic acid-induced neurotoxicity via maintaining NGF signalling pathway.
Authors claim that piperine protects hippocampal neurons against KA-induced exci-totoxicity by upregulating the NGF/TrkA/Akt/GSK3β signalling pathways.
any other behavioral parameters should have given better insights into the effects of piperidine
Author Response
We thank the reviewer for the critical comments and constructive suggestions.
Authors claim that piperine protects hippocampal neurons against KA-induced exci-totoxicity by upregulating the NGF/TrkA/Akt/GSK3β signalling pathways.
any other behavioral parameters should have given better insights into the effects of piperidine
As suggestion by the reviewer, the sentence is modified to 「improve depression-like behavior and cognitive impairment (Page 2, Line 55-56).
In the present study we focused on the anti-seizure activity of piperine. Thus, we only examined the effect of piperine on the seizure behavior. We agree the mentioned by the reviewer that other behavioral tests are required to perform. However, we were unable to complete the reviewer's recommendation due to the lack of other behavioral experimental equipment. Hope reviewer can accept our reply.